# Towards Universal Stimuli-Responsive Drug Delivery Systems: Pillar[5]arenes Synthesis and Self-Assembly into Nanocontainers with Tetrazole Polymers

**DOI:** 10.3390/nano11040947

**Published:** 2021-04-08

**Authors:** Dmitriy N. Shurpik, Lyaysan I. Makhmutova, Konstantin S. Usachev, Daut R. Islamov, Olga A. Mostovaya, Anastasia A. Nazarova, Valeriy N. Kizhnyaev, Ivan I. Stoikov

**Affiliations:** 1A. M. Butlerov Chemical Institute, Kazan Federal University, Kremlevskaya, 18, 420008 Kazan, Russia; dnshurpik@mail.ru (D.N.S.); lays_9393@mail.ru (L.I.M.); olga.mostovaya@mail.ru (O.A.M.); anas7tasia@gmail.com (A.A.N.); 2Institute of Fundamental Medicine and Biology, Kazan Federal University, Kremlevskaya, 18, 420008 Kazan, Russia; k.usachev@mail.ru; 3FRC Kazan Scientific Center, Russian Academy of Sciences, Arbuzov Institute of Organic and Physical Chemistry, Arbuzov St., 8, 420088 Kazan, Russia; daut1989@mail.ru; 4Department of Theoretical and Applied Organic Chemistry and Polymerization Processes, Irkutsk State University, K. Marksa, 1, 664003 Irkutsk, Russia; kizhnyaev@chem.isu.ru

**Keywords:** pillar[5]arene, tetrazole, drug delivery systems, fluorescein

## Abstract

In this work, we have proposed a novel universal stimulus-sensitive nanosized polymer system based on decasubstituted macrocyclic structures—pillar[5]arenes and tetrazole-containing polymers. Decasubstituted pillar[5]arenes containing a large, good leaving tosylate, and phthalimide groups were first synthesized and characterized. Pillar[5]arenes containing primary and tertiary amino groups, capable of interacting with tetrazole-containing polymers, were obtained with high yield by removing the tosylate and phthalimide protection. According to the fluorescence spectroscopy data, a dramatic fluorescence enhancement in the pillar[5]arene/fluorescein/polymer system was observed with decreasing pH from neutral (pH = 7) to acidic (pH = 5). This indicates the destruction of associates and the release of the dye at a pH close to 5. The presented results open a broad range of opportunities for the development of new universal stimulus-sensitive drug delivery systems containing macrocycles and nontoxic tetrazole-based polymers.

## 1. Introduction

In recent years, the pharmaceutical industry has had an increasing interest in the development and methods of introducing nanosystems in the treatment of various diseases by encapsulating drugs in biocompatible polymer matrices [1,2,3]. The resulting polymer-drug associates can change the pharmacokinetic properties and profile of the loaded drug after administration and provide a controlled and long-term effect of drugs on disease foci in comparison with the effect of the drug itself [1,2,3]. In addition, the polymer shell protects the loaded drug from premature biotransformation and can transport the drug to the focus of the disease practically without damage [1]. Water-soluble polymer systems occupy a special place among such drug delivery systems (DDSs) [4,5]. Various types of synthetic and natural polymer compositions (solid–liquid nanoparticles, liposomes, etc.) are widely studied as promising DDSs [6,7]. However, polymer systems have a number of disadvantages, namely, an extremely developed spatial structure and weak receptor properties, which complicate their controlled interaction with drugs [8,9].

The most promising DDSs are amphiphilic copolymers since their use promotes the solubilization of slightly soluble drugs [10]. It should be noted that the application of aqueous polymer systems is complicated due to the uncontrolled processes of association and aggregation of high molecular weight polymer units [11]. To address the problem, it was proposed to use various multifunctional macrocyclic compounds capable of controlled interaction with the polymer, which leads to the formation of stable associates in water [12]. Today, the most promising macrocyclic compounds are representatives of a new class of para-cyclophanes—pillar[n]arenes [13,14,15]. Pillar[n]arenes are macrocyclic compounds in which fragments of substituted hydroquinones are interconnected by methylene bridges [13]. Unlike other classes of macrocycles (calix[n]arenes, cyclodextrins, cucurbit[n]urils), pillar[n]arenes [16] are synthetically available and allow working in conditions (pH, water, and buffer systems) that are not suitable for other macrocycles [17,18,19,20,21,22]. Yang et al. [23] used a water-soluble pillar[5]arene to increase biocompatibility in antibacterial polymeric materials based on cationic polyaspartamide derivatives with different side-chain lengths by creating host–guest complexes to produce new antibacterial materials with pH-sensitive characteristics. Tong et al. [24] showed stimulus-responsive polymer vesicles consisting of water-soluble pillar[5]arenes with carboxylate fragments and paraquat-containing block copolymers in water. Vesicles are formed due to the formation of inclusion complexes between the pillar[5]arene and fragments of paraquats included in the structure of polymers. Additionally, Pisagatti et al. [25] developed a new type of coating based on pillar[5]arene 1 containing carboxylate fragments (Figure 1), with poly(allylamine hydrochloride) for slow release of antibiotics against Gram-positive and Gram-negative bacteria with antiadhesive and antimicrobial activity in vitro.

In this study, we propose a new type of universal stimulus-sensitive DDS based on decasubstituted macrocyclic structures—pillar[5]arenes and tetrazole-containing polymers. The versatility of the system lies in the principle of step-by-step supramolecular self-assembly of DDS components. The macrocyclic compound in this case will act as a link between the “protective shell”—a nontoxic water-soluble polymer and a drug. This is due to the presence of a macrocyclic cavity in pillar[5]arenes [17,18,19,20,21,22], which is involved in the formation of host-guest complexes with drugs of various structures [18,21,22]. Additionally, the introduction into the structure of macrocycles of substituents complementary to the tetrazole fragments of the polymer and sensitive to pH changes will facilitate the packing of tetrazole-containing polymers into nanosized associates [17,19].

Recently, polymer compositions based on polyvinyltetrazoles (PVTs) (Figure 1) are considered promising carriers for the formation of DDSs [26,27,28,29]. Polymers based on PVTs are easily synthesized, have a pronounced anti-inflammatory activity, promote blood clotting, accelerate wound healing [30,31]. One of the key conditions for the creation of DDSs based on water-soluble polymers is the micellization of polymer systems in a wide pH range [32,33,34]. DDSs based on polymer systems are of practical interest as nanocarriers for the encapsulation and controlled release of hydrophobic drugs. Such nanosized polymer systems are sensitive to the ambient pH due to the presence of charged ammonium fragments in their structure [33]. Tetrazole-containing polymer systems are ideal as components of pH-sensitive DDSs. However, it should be noted that polymers based on PVTs do not form stable nanosized aggregates in aqueous solutions [35]. In this regard, the formation of stable, stimulus-responsive nanoscale associates of PVT/pillararene capable of controlled drug release is an actual problem.

Thus, in this paper, we report the first example of the use of uncharged water-soluble derivatives of pillar[5]arene containing tertiary amino groups to obtain stimulus-responsive nanosized associates with polyvinyl (tetrazol-2-yl) ethyl ether (PVTE) and dye—fluorescein, by the method of controlled supramolecular self-assembly.

## 2. Materials and Methods

### 2.1. Characterization

^1^H, ^13^C NMR, and 2D NOESY, HSQC NMR spectra were obtained on a Avance-400 spectrometer (Bruker, Billerica, MA, USA) (^13^C{^1^H}—100 MHz and ^1^H and 2D NOESY—400 MHz). Chemical shifts were determined against the signals of residual protons of deuterated solvent (CDCl_3_, D_2_O, CD_3_OD). The concentration of sample solutions was 3–5%.

Attenuated total internal reflectance IR spectra were recorded with a Spectrum 400 (Perkin Elmer, Waltham, MA, USA) Fourier spectrometer.

Elemental analysis was performed with a Perkin Elmer 2400 Series II instrument (Perkin Elmer, Waltham, MA, USA).

Mass spectra (MALDI-TOF) were recorded on an Ultraflex III mass spectrometer (Bruker, Billerica, MA, USA) in a 4-nitroaniline matrix. Melting points were determined using a Boetius Block apparatus.

Additional control of the purity of compounds and monitoring of the reaction were carried out by thin-layer chromatography using Silica G, 200 µm plates, UV 254.

Most chemicals were purchased from Aldrich and used as received without additional purification. Organic solvents were purified in accordance with standard procedures.

### 2.2. Diffusion Ordered Spectroscopy (DOSY)

^1^H diffusion ordered spectroscopy (DOSY) spectra were recorded on a Bruker Avance 400 spectrometer (Bruker, Billerica, MA, USA) at 9.4 tesla at a resonating frequency of 400.17 MHz for ^1^H using a BBO (Bruker, Billerica, MA, USA) 5 mm gradient probe. The temperature was regulated at 298 K and no spinning was applied to the NMR tube. DOSY experiments were performed using the STE bipolar gradient pulse pair (stebpgp1s) pulse sequence with 16 scans of 16 data points collected. The maximum gradient strength produced in the z direction was 5.35 G mm^−1^. The duration of the magnetic field pulse gradients (δ) was optimized for each diffusion time (Δ) in order to obtain a 2% residual signal with the maximum gradient strength. The values of δ and Δ were 1.800 μs and 100 ms, respectively. The pulse gradients were incremented from 2 to 95% of the maximum gradient strength in a linear ramp.

### 2.3. Scanning Electron Microscope (SEM)

Morphological structures of samples were observed by SEM (Carl Zeiss Auriga Cross Beam). Samples were first diluted with water to a final concentration of 1 × 10^−4^ g/mL, and the resulting suspension was placed on a silicon pan, which was then dried in a vacuum desiccator for 1 h. After complete drying, the samples were placed into the scanning electron microscope using a special holder for microanalysis. Analysis was held at the accelerating voltage of 80 kV.

### 2.4. Fluorescence Spectroscopy

Fluorescence spectra were recorded on a Fluorolog 3 luminescent spectrometer (Horiba Jobin Yvon, Longjumeau, France). The excitation wavelength was selected as 450 nm. The emission scan range was 480–700 nm. Excitation and emission slits were 1 nm. Quartz cuvettes with an optical path length of 10 mm were used. Fluorescence spectra were automatically corrected by the Fluorescence program. The spectra were recorded in water solutions with a concentration of fluorescein 1 × 10^−5^ M for pillar[5]arene/fluorescein systems. The obtained molar ratio of fluorescein to pillar[5]arene 7 was 1:100. The experiment was carried out at 293 K. Fluorescence spectra at different pH were fixed in a buffer for system **7** (1 × 10^−5^ M)/Flu (1 × 10^−5^ M)/PVTE (1 × 10^−4^ M) at 298 K. Solutions of the investigated systems were measured after incubating for an hour at room temperature.

### 2.5. UV–Visible Spectroscopy

UV–Vis spectra were recorded using the UV-3600 spectrometer (Shimadzu, Kyoto, Japan); the cell thickness was 1 cm, and slit width 1 nm. Deionized water with a resistivity >18.0 MΩ cm was used to prepare the solutions. Deionized water was obtained from a Millipore-Q purification system. Recording of the absorption spectra of the mixtures of PVT, PVTE, and fluorescein (Flu) with pillar[5]arenes **7–9** and **12** (1 × 10^−5^–5 × 10^–5^ M) was carried out after mixing the solutions at 293 K. The 1 × 10^−5^–5 × 10^–5^ M solution of the guest—PVT, PVTE (50, 75, 100, 150, 300, 400, 500, 600, 700, 900, 1200, 1500, 1800, 2100, 2400, and 2700 µL, 1 × 10^−5^–5 × 10^−5^ M) in deionized water or phosphate buffer—was added to 300 µL of the solution of macrocycle **7–9** and **12** (1 × 10^−5^–5 × 10^–5^ M) (in the case of Flu, the concentration of Flu was constant) in deionized water or phosphate buffer and diluted to a final volume of 3 mL with deionized water or phosphate buffer. The UV spectra of the solutions were then recorded. The association constants of complexes were calculated as described below. Three independent experiments were carried out for each series. The student’s *t*-test was applied in the statistical data processing.

### 2.6. Dynamic Light Scattering (DLS)

#### 2.6.1. Particle Size

Dynamic Light Scattering (DLS). The particle size was determined by the Zetasizer Nano ZS (Malvern Instruments, Malvern, UK) at 298 K. The instrument contains a 4 mW He—Ne laser operating at a wavelength of 633 nm and incorporated noninvasive backscatter (NIBS) optics. The measurements were performed at the detection angle of 173° and the software automatically determined the measurement position within the quartz cuvette. The concentration ratios of pillar[5]arenes 7–9 and 12, pillar[5]arenes **7–9** and **12**/PVT, PVTE, and Flu were 50:1, 10:1, 5:1, 2:1, 1:1, 1:2, 1:5, and 1:15, and the concentration of compounds was 1 × 10^−5^ M, 1 × 10^−4^ M, and 1 × 10^−3^ M. The experiments were carried out for each solution in triplicate.

#### 2.6.2. Zeta Potentials

Zeta (ζ) potentials were measured on a Zetasizer Nano ZS from (Malvern Instruments, Malvern, UK). Samples were prepared as they were for the DLS measurements and were transferred with the syringe to the disposable folded capillary cell for measurement. The zeta potentials were measured using the Malvern M3-PALS method and averaged from three measurements.

### 2.7. X-ray Diffraction (XRD)

Data sets for single crystals **6**, **7,** and **11** were collected on a Rigaku XtaLab Synergy S instrument with a HyPix detector (Rigaku, Tokyo, Japan) and a PhotonJet microfocus X-ray tube (Rigaku, Tokyo, Japan) using Cu Kα (1.54184 Å) radiation at low temperature. Images were indexed and integrated using the CrysAlisPro data reduction package. Data were corrected for systematic errors and absorption using the ABSPACK module, i.e., numerical absorption correction based on Gaussian integration over a multifaceted crystal model and empirical absorption correction based on spherical harmonics according to the point group symmetry using equivalent reflections. The GRAL module was used for the analysis of systematic absences and space-group determination. The structures were solved by direct methods using SHELXT [36] and refined by the full-matrix least squares on F^2^ using SHELXL [37]. Non-hydrogen atoms were refined anisotropically. The hydrogen atoms were inserted at the calculated positions and refined as riding atoms. The positions of the hydrogen atoms of methyl groups were found using rotating group refinement with idealized tetrahedral angles. The figures were generated using Mercury 4.1 [38] program. To refine the model against the measured data of **11** the SQUEEZE as implemented in PLATON was used to model the disordered solvent in the big voids of the structure. SQUEEZE identified a void with a volume of 2190 Å^3^ containing the equivalent of 768 electrons. This would correspond to about 3[CHCl_3_ + C_2_H_3_N]. Crystal data, data collection, and structure refinement details are summarized in Appendix A.

### 2.8. Synthesis of O-Substituted Hydroquinones and PVT and PVTE Polymers

Compound **5**, **10** were synthesized according to the literature procedures [39,40].

Polymers PVT and PVTE were synthesized according to the literature procedures [35].

Polymer sample PVT (M = 7 × 10^4^) used in this study was synthesized via the azidation of polyacrylonitrile, which was synthesized through the 2,2’-azodi(isobutyronitrile) (AIBN)-initiated polymerization of 5-Vinyl-1H-tetrazole (VT) in acetonitrile at 60 °C.

We used a PVTE polymer sample (M = 5 × 10^4^) synthesized by cyanoethylation of polyvinyl alcohol with acrylonitrile, followed by azidation of nitrile groups.

### 2.9. Synthesis

#### 2.9.1. General Procedure for the Synthesis of Compounds **6** and **11**

Paraformaldehyde (PFA) 0.17 g (5.9 mmol) and 50 mL of dichloroethane were placed in a round bottom flask equipped with a magnetic stirrer. Then, (1.9 mmol) of compound **5** (or **10**) and 0.17 mL (1.9 mmol) of trifluoromethanesulfonic acid were added. The reaction was carried out at 0 °C for 30 min. Then, the temperature was gone up to 45 °C and stirred for 1 h. Then, the reaction mixture was quenched with 5% NaHCO_3_ solution (50 mL). The organic layer was separated on a separatory funnel and washed with distilled water (2 × 30 mL) and then evaporated under reduced pressure. The resulting light brown reaction mixture was purified by reprecipitation from isopropyl alcohol. The reaction mixture was dissolved in acetonitrile (20 mL) and slowly poured into isopropyl alcohol (40 mL). The formed precipitate was separated by centrifugation and dried at room temperature.


*4,8,14,18,23,26,28,31,32,35-Deca-(4-methylbenzylsulfonate-1-ethoxy)-pillar[5]arene (**6**).*


Light yellow crystalline substance. Yield 0.74 g (85%). ^1^H NMR (400 MHz, CDCl_3_): δ 2.27 (s., 30H, -CH_3_), 3.60 (s., 10H, -CH_2_-), 4.02–4.11 (m., 10H, -O-CH_2_-CH_2_-O-), 4.21–4.29 (m., 10H, -O-CH_2_-CH_2_-O-), 4.29–4.38 (m., 10H, -O-CH_2_-CH_2_-O-), 4.40–4.53 (m., 10H, -O-CH_2_-CH_2_-O-), 6.78 (s, 10H, ArH), 7.05 (d., J = 7.9 Hz, 20H, Ar_Ts_CH_3_), 7.71 (d., J = 7.9 Hz, 20H, Ar_Ts_CH_3_). ^13^C NMR (100 MHz, CDCl_3_): δ 21.62, 28.96, 66.01, 69.85, 114.50, 127.98, 128.55, 130.01, 132.28, 144.98, 149.49. IR, ν/cm^−1^: 3065 (ArH), 2927, 2877 (-CH_2_-, CH_3_), 1597, 1496 (-C=C-), 1353, 1172 (-S=O), 917 (-S-O). MS (MALDI-TOF) calc. [M^+^] *m/z* = 2591.5, found [M^+^] *m*/*z* = 2591.7. Found (%): C, 56.12; H, 5.25; S, 10.93. Calc. for C_125_H_130_O_40_S_10_. (%): C, 57.90; H, 5.05; S 12.36.


*4,8,14,18,23,26,28,31,32,35-Deca-[(isoindoline-1,3-dione)propoxy]-pillar[5]arene (**11**).*


Light yellow crystalline substance. Yield 0.70 g (74%). ^1^H NMR (400 MHz, CDCl_3_): δ 2.47–2.63 (m., 10H, -O-CH_2_-CH_2_-CH_2_-N), 4.10 (s., 10H, -CH_2_-), 4.22–4.32 (m., 10H, -O-CH_2_-CH_2_-CH_2_-N), 4.32–4.50 (m., 20H, -O-CH_2_-CH_2_-CH_2_-N), 7.26 (s., 10H, ArH), 7.97–8.21 (Ar_Pht_H). ^13^C NMR (100 MHz, CDCl_3_): δ 29.26, 29.43, 35.81, 66.35, 115.28, 123.25, 128.53, 132.40, 133.84, 149.95, 168.31. IR, ν/cm^−1^: 3029 (ArH), 2941, 2878 (-CH_2_-, CH_3_), 1703 (-C=O), 1203 (Ar-O-CH_2_-). MS (MALDI-TOF) calc. [M^+^] *m/z* = 2481.8, found [M+H^+^] *m*/*z* = 2484.7. Found (%): C, 68.96; H, 4.46; N, 4.84. Calc. for C_145_H_120_N_10_O_30_. (%): C, 70.15; H, 4.87; N, 5.64.

#### 2.9.2. General Procedure for the Synthesis of Compounds **7**–**9** and **12**

Compound **6** (0.2 g, 0.07 mmol) and 1 mL amine (pyrrolidine, piperidine, morpholine) were placed in a round-bottom flask with a magnetic stirrer and a reflux condenser. The reaction mixture was boiled for 48 h. Then, the dark brown reaction mixture was poured into diethyl ether (10 mL). The formed precipitate was collected by filtration on a Schott filter. Then, the precipitate was dissolved in dichloromethane (20 mL) and washed with distilled water (3 × 20 mL). The organic layer was separated and evaporated under reduced pressure. The target products were obtained by recrystallization from isopropyl alcohol.


*4,8,14,18,23,26,28,31,32,35-Deca-[2-(pyrrolidin-1-yl)ethoxy]-pillar[5]arene (**7**).*


White crystalline substance. Yield 0.11 g (87%). ^1^H NMR (400 MHz, CDCl_3_): δ 1.70–1.90 (m., 40H, -N-(CH_2_)_2_-), 2.55–2.77 (m., 40H, -N-(CH_2_)_2_-), 2.87–3.05 (m., 20H, -O-CH_2_-CH_2_-), 3.75 (s., 10H, -CH_2_-), 3.93–4.16 (m., 20H, -O-CH_2_-CH_2_-), 6.85 (s., 10H, ArH). ^13^C NMR (100 MHz, CDCl_3_): δ 22.88, 28.97, 54.57, 54.95, 66.96, 117.26, 124.41, 149.66. IR, ν/cm^−1^: 3029 (ArH), 2954, 2882 (-CH_2_-), 2607, 2483 (-N-(CH_2_)_2_-), 1204 (Ar-O-CH_2_-). MS (MALDI-TOF) calc. [M^+^] *m/z* = 1582.1, found [M+H^+^] *m*/*z* = 1583.2. Found (%): C, 71.20; H, 7.98; N, 7.74. Calc. for C_95_H_140_N_10_O_10_. (%): C, 72.12; H, 8.92; N, 8.85.


*4,8,14,18,23,26,28,31,32,35-Deca-[2-(piperidin-1-yl)ethoxy]-pillar[5]arene (**8**).*


Light yellow crystalline substance. Yield 0.10 g (78%). ^1^H NMR (400 MHz, CDCl_3_): δ 1.33–1.54 (m., 20H, -CH_2_-), 1.54–1.70 (m., 40H, -N-(CH_2_)_2_-(CH_2_)_2_-), 2.50–2.70 (m., 40H, -N-(CH_2_)_2_-(CH_2_)_2_-), 2.70–2.91 (m., 20H, -O-CH_2_-CH_2_-), 3.73 (s., 10H, -CH_2_-), 3.90–4.14 (m., 20H, -O-CH_2_-CH_2_-), 6.83 (s., 10H, ArH). ^13^C NMR (100 MHz, CDCl_3_): δ 24.34, 26.15, 29.48, 58.68, 66.81, 115.19, 128.53, 149.91. IR, ν/cm^−1^: 2929, 2851 (-CH_2_-), 2781, 2748 (-N-(CH_2_)_2_-), 1203 (Ar-O-CH_2_-). MS (MALDI-TOF) calc. [M^+^] *m/z* = 1722.2, found [M+H^+^] *m*/*z* = 1723.1. Found (%): C, 72.54; H, 8.93; N, 7.86. Calc. for C_105_H_160_N_10_O_10_. (%): C, 73.22; H, 9.36; N, 8.13.


*4,8,14,18,23,26,28,31,32,35-Deca-(2-morpholinoethoxy)-pillar[5]arene (**9**).*


Light yellow crystalline substance. Yield 0.09 g (72%). ^1^H NMR (400 MHz, CDCl_3_): δ 2.60 (d., 40H, J = 4.4 Hz, -N-(CH_2_)_2_-(CH_2_)_2_-O-), 2.75–2.90 (m., 20H, -O-CH_2_-CH_2_-), 3.69–3.77 (m., 50H, -N-(CH_2_)_2_-(CH_2_)_2_-O-, -CH_2_-), 3.92–4.14 (m., 20H, -O-CH_2_-CH_2_-), 6.84 (s, 10H, ArH). ^13^C NMR (100 MHz, CDCl_3_): δ 29.59, 54.09, 54.33, 58.28, 66.76, 67.03, 115.73, 128.95, 150.03. IR, ν/cm^−1^: 2935, 2891 (-CH_2_-), 2800 (-N-(CH_2_)_2_-), 1113 (Ar-O-CH_2_-). MS (MALDI-TOF) calc. [M^+^] *m/z* = 1742.0, found [M+H^+^] *m*/*z* = 1743.0. Found (%): C, 66.35; H, 9.25; N, 8.57. Calc. for C_95_H_140_N_10_O_20_. (%): C, 65.49; H, 8.10; N, 8.04.

4,8,14,18,23,26,28,31,32,35-Deca-(aminopropyloxy)-pillar[5]arene (**12**).

Macrocycle **11** (0.3 g, 0.12 mmol), hydrazine hydrate (65% aqueous solution, 0.53 mL, 7.2 mmol), and 10 mL of methanol were placed in a round bottom flask with a magnetic stirrer and a reflux condenser. The reaction mixture was boiled for 48 h. Then, the reaction mixture was cooled to room temperature. The formed precipitate was filtered off on a Schott filter. The crude product was washed on the filter with methanol excess. Then, the resulting product was dissolved in 10% HCl (15 mL). The resulting cloudy solution was filtered on a Schott filter. Next, an ammonia solution (15%) was added to the filtrate to pH 10. The precipitated amine **12** was filtered off on a Schott filter. The resulting white crystalline substance was dried under vacuum. Yield 0.12 g (85%). ^1^H NMR (400 MHz, D_2_O): δ 1.90–2.06 (m., 20H, -O-CH_2_-CH_2_-CH_2_-NH_2_), 3.15 (t., 20H, J = 7.3 Hz, -O-CH_2_-CH_2_-CH_2_-NH_2_), 3.78–3.90 (m., 30H, -O-CH_2_-CH_2_-CH_2_-NH_2_, -CH_2_-), 6.76 (s, 10H, ArH). ^13^C NMR (100 MHz, CDCl_3_): δ 26.78, 29.50, 37.31, 67.14, 116.61, 129.78, 150.05. IR, ν/cm^−1^: 3298 (-NH_2_), 2924, 2864 (-CH_2_-), 1199 (Ar-O-CH_2_-). MS (MALDI-TOF) calc. [M^+^] *m/z* = 1180.8, found [M+H^+^] *m*/*z* = 1181.6. Found (%): C, 65.93; H, 7.45; N, 10.84. Calc. for C_65_H_100_N_10_O_10_. (%): C, 66.07; H, 8.53; N, 11.85.

Detailed information on physical–chemical characterization is presented in Appendix A.

## 3. Results

### 3.1. Synthesis of Pillar[5]arene Derivatives

The two most studied tetrazole-containing polymers were selected as objects of this study, namely, poly-5-vinyltetrazole (PVT) and polyvinyl (tetrazol-5-yl) ethyl ether (PVTE) (Figure 1), which have relatively low toxicity (from LD_50_ = 166 mg/kg for p-5VT to 1340 mg/kg for IPT-VPD copolymer) [41]. Interest in tetrazole-containing polymers is growing due to the fact that the tetrazole fragment in the polymer structure is a pharmacophore group. Today, there are 43 drugs containing 1H- or 2H-tetrazole, 23 of them are approved by the FDA [42]. Moreover, 5-substituted tetrazole is of particular interest because it has a mobile proton. This makes 5-substituted tetrazoles strong NH acids (pKa = 4.5–4.9 depending on the substituent). Therefore, 5-substituted tetrazoles have been widely used in recent years as an isosteric analog of the COOH group [42,43,44]. Similar to polymers containing carboxylate groups, polymers containing tetrazole moieties (PVT and PVTE) are capable of ionizing at physiological pH (7.4). However, it has been shown [43] that tetrazolate anions are almost 10 times more lipophilic than carboxylate anions, which is an important factor in the passage of the polymer composition through cell membranes. In addition, the polymers used in DDSs and containing carboxylate groups (*N*-isopropylacrylamide (NIPAM), poly(alkyl acrylic acid)s, modified poly(glycidol)s (PGs)) do not penetrate into the cell due to the submicron size, but they are fixed on the surface [44]. Delivery of the drug into the cell is carried out by loosening the cell membrane with subsequent release of the drug [44]. Loosening of the cell membrane during drug transport can be detrimental to the cell. The interaction between water-soluble polymers and lipid membranes is mainly governed by the hydrophobic/hydrophilic balance and the polymer architecture [45]. Charged polymers (polyelectrolytes), such as polycarboxylates or polyamines, can adsorb on oppositely charged lipid bilayers, which can lead to membrane deformations. This manifests itself in an increase in the curvature of the membrane, which can lead to sealing or destruction of the membrane in the case of cationic polymers [45,46]. Membrane damage occurs through hole formation, and some cationic polymers/oligomers can move across the cell membrane and bind to intracellular DNA or RNA to prevent intracellular synthesis [46]. In previous works, we showed [47] that polymers based on tetrazole in the presence of pillar[5]arene are capable of forming nanoscale associates. In [47], uncharged pillar[5]arenes containing hydroxyl groups were used. We have shown that the macrocyclic platform plays a key role in the formation of supramolecular associates. The resulting nanosized associates turned out to be insensitive to pH changes. Therefore, in this work, it was decided to use pillar[5]arenes containing primary and tertiary amino groups capable of protonation in an acidic medium.

In this regard, we hypothesize that the use of water-soluble derivatives of the pillar[5]arene containing amino groups in the presence of tetrazole-containing polymers will contribute to the formation of nanoscale associates sensitive to changes in the pH environment. Additionally, the presence of a free electron-donating macrocyclic cavity makes it possible to use substituted pillar[5]arenes as molecular containers. To confirm this hypothesis, it was necessary to solve a number of problems, i.e., (1) to synthesize macrocycles with functional groups complementary to tetrazole fragments and capable of interacting with the polymer, (2) to study the interaction of synthesized macrocycles with polymers (PVT and PVTE), and (3) to study the interaction of macrocycles and polymers with the model drug. The key task of the study is to establish the patterns of drug release when the system is exposed to a certain stimulus. The following functional groups of pillar[5]arene can act as groups interacting with the polymer: -NH (aliphatic or aromatic), -OH (aliphatic, phenolic, aromatic), and carbonyl (esters, amides, ketones, etc.) groups [42,43,44]. Analysis of the literature has shown that four main pillar[5]arenes **1–4** (Figure 1) are the most studied as components of self-assembling systems [48,49,50,51,52,53]. The presence of acidic protons in the tetrazole fragments of PVT and PVTE (Figure 1) excludes the use of carboxylate groups (compounds **1**, **2** in Figure 1) in the pillar[5]arene structure for further study of the interaction with polymers. For this reason, we chose macrocycles **3** and **4**. However, the method for their preparation requires the use of an expensive palladium catalyst and gaseous compounds (hydrogen and trimethylamine) [52,53].

In order to develop an alternative approach to the production of pillar[5]arenes containing primary and tertiary amino acids, the synthesis of the pillar[5]arene platform was carried out by macrocyclization of functionalized 1,4-dialkoxybenzenes: *p*-bis[2-(phthalimido)ethyloxy]benzene and 2,2′-bis(tosylethyl)-hydroquinone. The cyclization of O-substituted hydroquinones in the presence of paraformaldehyde is currently the main method for the preparation of substituted pillar[n]arenes. However, the yield of the anticipated pillararenes modified with functional groups is quite low [16]. Therefore, the development of additional methods of macrocyclization makes it possible to significantly expand the number of introduced functional groups into the pillar[5]arene structure.

The starting bis[2-(phthalimido)ethyloxy]benzene **5** and 2,2′-bis(tosylethyl)-hydroquinone **10** were obtained according to published methods [39]. At the next stage, we selected the conditions for macrocyclization (see Appendix A). Three main solvents were selected in the synthesis of pillar[n]arenes: CH_2_Cl-CH_2_Cl, CHCl_3_, CH_2_Cl_2_ [13,14,15,16]. Additionally, catalysts (Lewis acids) were varied. A number of main catalysts were selected, namely, BF_3_×Et_2_O, AlBr_3_, CF_3_COOH, and CF_3_SO_3_H. These catalysts are most often used in the synthesis of pillar[5]arenes [13,14,15,16]. Paraform and trioxane were chosen as a reagent, allowing the introduction of methylene bridge into the macrocycle. The reaction time varied from 30 min to 24 h. The reaction temperature range was from 0 °C to 85 °C (Scheme 1) (see Appendix A).

The change in the nature of the solvent showed that the reaction of **5** and **10** with paraformaldehyde or trioxane in anhydrous CH_2_Cl_2_ in the presence of Lewis acids (BF_3_×Et_2_O, AlBr_3_, CF_3_COOH, CF_3_SO_3_H) does not lead to the formation of desired products **6** and **11**. In the case of CHCl_3_, macrocyclization proceeded only with 2,2′-bis(tosylethyl)-hydroquinone in the presence of paraformaldehyde and CF_3_SO_3_H. Analysis of ^1^H NMR spectra showed the presence of a mixture of products—pillar[5]arene, pillar[6]arene, and polymer (see Appendix A). Unfortunately, it was not possible to separate the reaction mixture using column chromatography. The use of CH_2_Cl-CH_2_Cl as a solvent and CF_3_SO_3_H as catalyst made it possible to obtain target macrocycles **6** and **11** (Scheme 1). The use of BF_3_×Et_2_O as catalyst also led to the production of **6** (64%) and **11** (53%)**.** However, a large amount of polymer was formed during the reaction, which required laborious column chromatography. The formation of target products was not observed in the case of using catalysts AlBr_3_ and CF_3_COOH. Changing the paraformaldehyde to trioxane in the BF_3_×Et_2_O/CH_2_Cl-CH_2_Cl and CF_3_SO_3_H/CH_2_Cl-CH_2_Cl systems also led to **6** and **11**, but the yield of the target products did not exceed 20%.

Thus, the use of model compounds **5** and **10** in the presence of paraformaldehyde and CF_3_SO_3_H leads to the formation of target macrocycles **6** and **11** in 85% and 74% yields, respectively (Scheme 1). The temperature regime in all performed reactions varied from 0–85 °C. However, the best results were achieved when the reaction was carried out in the temperature range from 0 °C to 45 °C for an hour and a half for both macrocycles. It should be noted that the formation of non-macrocyclic polymer is almost minimized when the reaction goes in the selected temperature range and use CF_3_SO_3_H as a catalyst. The target products were isolated by reprecipitation from isopropyl alcohol. The technique developed by us allows producing large quantities of target macrocycles without losing the yield (see Appendix A).

The spatial structure of the resulting products **6** and **11** (see Appendix A.) was fully confirmed using structural analysis by single-crystal XRD (Figure 2). The crystals were grown in both cases from a mixture of solvents CHCl_3_-CH_3_CN. Syngony of product **6** is monoclinic, group symmetry is C2/c (Figure 2a,b). As can be seen from Figure 2c, the formation of a supramolecular polymer in the crystalline state is observed in the case of compound **6** by including two tosylate fragments in the cavity of a single macrocycle.

The type of crystal lattice is triclinic, and the symmetry group is *P-*1 in the case of macrocycle **11** (Figure 3). Unlike macrocycle **6**, compound **11** does not form a supramolecular polymer in the crystalline state. All phthalimide fragments **11** (Figure 3a,b) are deployed in the “propeller blade” form outward from the cavity of the macrocycle **11**. This arrangement can be explained by intermolecular π–π-stacking interactions between phthalimide fragments of neighboring macrocycles in structural motive **11**. Structural analysis by single-crystal XRD showed the presence of pseudocavity (Figure 3c) in the crystal structure of **11**, which is formed between two macrocycles bound by stacking phthalimide fragments (Figure 3c, orange balls). It is interesting to note that the absence of substituent fragments in the cavity of the macrocyclic platform **7** (Figure 3c, green balls) makes it possible to use the resulting structures as nonporous adaptive crystals [54,55,56,57,58].

Then, macrocycle **6** was involved in the reaction with secondary amines—pyrrolidine, piperidine, and morpholine. The reaction was performed in excess of amine at 100 °C (Scheme 1). The reaction time was 48 h. The yield of macrocycles **7–9** was from 72% to 87% (see Appendix A). In order to obtain a primary amine, macrocycle **11** was involved in a reaction with hydrazine hydrate in methanol at the boiling point of the solvent (Scheme 1). The reaction time was 48 h. Decaamine **12** precipitated from the reaction mixture was filtered off and purified by conversion into the hydrochloride form. The yield of macrocycle **12** was 85%. The structures of the obtained products **7–9** and **12** were fully confirmed by a complex of physical methods: ^1^H and ^13^C one-dimensional NMR, IR spectroscopy, and mass spectrometry, and the compositions were confirmed by elemental analysis data (see Appendix A). The structure of product **7** was finally established by XRD (Figure 4). The type of crystal lattice is triclinic, symmetry groups are *P*-1 (Figure 4a). As can be seen from Figure 4b, the crystal structure of macrocycle **7** is a hollow tube in which the macrocycle molecules are arranged one after the other. It should be noted that such packing of macrocycles into hollow tubes (Figure 4b, yellow balls) makes these compounds promising for the development of crystalline organic materials based on macrocycles [56,57,58].

### 3.2. Study of the Interaction of Pillar[5]arenes with PVT and PVTE Polymers

At the next stage of the study, the ability of macrocycles **7–9** and **12** to interact with PVT, PVTE, and fluorescein dye in ethanol and water was studied using UV–Vis, ^1^H, 2D NMR, fluorescence spectroscopy, and dynamic light scattering (DLS). Unfortunately, changes in the UV–Vis spectra of compounds **7**–**9** and **12** in the presence of PVT and PVTE turned out to be insignificant in water and buffer solutions (pH = 5–9). It was found that the spectral changes were significant only during the interaction of macrocycles **7** and **9** with PVTE in ethanol, which made it possible to establish the quantitative characteristics of the binding (see Appendix A). The association constant was determined by spectrophotometric titration, in which the concentration of PVTE was changed, and the concentration of **7** (1 × 10^−5^ M) remained constant. The results were processed using BindFit [59,60,61] and fitted to a 1:1 binding model (see Appendix A). To confirm the proposed stoichiometry, the titration data were also processed using a binding model with a host:guest ratio of 1:2. However, in this case, the constants were determined with much greater uncertainty (see Appendix A). Calculated association constant (Ka) was for **9**/PVTE = 1898.15 and for **7**/PVTE—Ka = 2993.95. The interaction of macrocycle **12** with PVT and PVTE is accompanied by the formation of a precipitate. The calculation of the association constant by UV-Vis spectroscopy is impossible. The spectral pattern, in this case, is complicated by light scattering, which is manifested in a significant rise of the baseline.

We chose 2D ^1^H-^1^H NOESY and 2D DOSY NMR spectroscopy methods to confirm the formation of a complex between macrocycles **7** and **9** with PVTE and its structure. (Figure 5). However, an analysis of the experimental data for mixtures **9/**PVTE and **7**/PVTE, obtained using ^1^H NMR spectroscopy, did not allow determining the nature of the interaction since the polymer signals were too broadened. In this regard, to confirm the interaction of pillar[5]arenes with polymer fragments, the corresponding monomer, 5-vinyltetrazole (5-VT) was chosen. Thus, in the 2D ^1^H-^1^H NOESY NMR spectrum of the **7**/5-VT (5 × 10^−3^ M) associate in CD_3_OD (Figure 5a), cross peaks were observed between the protons of the 5-VT **H^3^** double bond and the protons of the pyrrolidine fragment **H^g^** and **H^f^** of macrocycle **7** (Figure 5c). The formation of the **7**/5-VT complex was additionally confirmed by 2D DOSY NMR spectroscopy (Figure 5b). Diffusion coefficients of **7**, 5-VT and **7**/5-VT at 298 K (1 × 10^−3^ M) were determined. The DOSY NMR spectrum of the **7**/5-VT system showed the presence of signals from a complex lying on one straight line (Figure 5b), with a diffusion coefficient (D = 3.36 × 10^−10^ m^2^ s^−1^), which is significantly lower than the self-diffusion coefficient of macrocycle **7** (D = 4.95 × 10^−10^ m^2^ s^−1^) and 5-VT (D = 7.44 × 10^−10^ m^2^ s^−1^) under the same conditions [21]. The results obtained unambiguously indicate the formation of the **7**/5-VT complex. The absence of cross peaks between 5-VT and aromatic protons of pillar[5]arene 7 indicates that the monomer is not in the macrocyclic cavity but in the pseudocavity of the macrocycle formed by fragments of substituents [62,63,64].

### 3.3. Study of the Interaction of Pillar[5]arenes with Fluorescein

The presence of a macrocyclic cavity in the pillar[5]arene 7 which is “not occupied” by a guest in the 7/5-VT system can facilitate the incorporation of an additional guest molecule into the pillararene cavity. To confirm this hypothesis, we studied the interaction of macrocycles **7** and **9** with a dye—fluorescein (Flu). Fluorescein was chosen as a convenient model compound. The choice of a specific drug for research is difficult since it is of interest to create universal DDS. Fluorescein is a well-studied compound with a developed spatial structure similar to a number of commercially used drugs: [65,66] quinoline derivatives (mepacrine, amodiaquine, etc.), which have antibacterial action, phenothiazine antipsychotics (aminazine, propazine, etc.), antineoplastic antibiotics (pixantrone, epirubicinum, etc.). A distinctive feature of fluorescein is the presence of its own fluorescence, which makes it possible to use sensitive fluorescent methods for establishing complexation processes.

As a result, the ability of macrocycles **7** and **9** to interact with fluorescein was studied using UV–Vis and fluorescence spectroscopy methods. Pillar[5]arenes 7 and 9 have one absorption maximum in ethanol with λmax at 292 nm according to UV–Vis spectroscopy data. Fluorescein solution absorbs at λmax = 276 nm, 453 nm, and 480 nm. Therefore, to study the interaction of macrocycles 7 and 9 with fluorescein, we chose the spectral range 350–600 nm, in which there is no absorption of pillar[5]arenes **7** and **9**. It turned out that the interaction of macrocycle **7** and **9** (1 × 10^−5^ M) with fluorescein leads to a hyperchromic effect, and the absorption band undergoes a redshift (see Appendix A). The association constant and the stoichiometry of the associate were determined by the spectrophotometric titration data, in which the concentration of macrocycles **7** and **9** varied at a constant concentration of fluorescein (1 × 10^−5^ M).

The processing of the results was based on the analysis of binding isotherms, for which the BindFit [59,60,61] application was used and fitted to a 1:1 binding model (see Appendix A). The association constant of pillar[5]arenes 7 and 9 with fluorescein was 4730 M−1 and 2699 M−1, respectively. Additionally, the stoichiometry of the complex was confirmed by titration data processed by the binding model at the ratio host-guest = 1:2 and 2:1. However, in this case, the constants are determined with great uncertainty. The binding constant of **7**/Flu in ethanol was additionally calculated by the method of fluorescence spectroscopy from analysis of binding isotherms. The dye in ethanol has maximum emission at 520 nm. The addition of pillararene **7** to it leads to the flare-up of fluorescence without a shift in the emission maximum. The association constant calculated using the 1:1 binding model in BindFit for this system is 10614.

At the next stage of the study, the self-association of macrocycles **7** and **9**, and the association with fluorescein and PVTE were studied using the dynamic light scattering (DLS) method. A self-association study of macrocycles **7** and **9** was carried out in water and ethanol using the DLS method (Appendix A). It turned out that pillar[5]arenes 7 and 9 do not form stable self-associates in ethanol over the entire studied concentration range (1 × 10^−3^–1 × 10^−5^ M). Further, the systems of macrocycle **7**/Flu and **9**/Flu were studied at the 1:1, 1:2, 2:1 ratios in the concentration range (1 × 10^−3^–1 × 10^−5^ M) (see Appendix A). However, stable associates are formed only when Flu is added to a solution of macrocyclic **7** in ethanol. The minimum values of the polydispersity index (PDI = 0.16) were recorded for the ratio **7**/Flu = 1:1 (1 × 10^−5^ M) with an average hydrodynamic diameter of 155 nm in ethanol (see Appendix A). An increase in the concentration of components **7**/Fly = 1:1 (1 × 10^−4^–1 × 10^−3^ M) leads to increasing the average hydrodynamic diameter of the formed associates. The largest value of the average hydrodynamic diameter of the **7**/Flu system at 1:1 ratio (D = 428 nm) is observed at 1 × 10^−3^ M concentration with PDI = 0.36. The **9**/Flu system at 1:1, 1:2, 2:1 ratios and in the studied concentration range (1 × 10^−3^–1 × 10^−5^ M) does not form stable associates in ethanol. It should be noted that both systems **7**/Flu and **9**/Flu do not form stable associates in aqueous and aqueous–alcoholic (H_2_O/C_2_H_5_OH = 100/1) solutions (1 × 10^−3^–1 × 10^−5^ M).

Then, the association of the **7**/PVTE and **9**/PVTE systems in ethanol and water was studied. PVTE does not form stable associates over the studied concentration range (1 × 10^−3^–1 × 10^−5^ M). The association of the **7**/PVTE and **9**/PVTE systems was studied at the 50:1, 10:1, 5:1, 2:1, 1:1, 1:2, 1:5, and 1:15 ratios in the concentration range (1 × 10^−3^–1 × 10^−5^ M). It is interesting that only in the case of macrocycle **7** (1 × 10^–4^ M) in the presence of PVTE (1 × 10^–5^ M) (**7**/PVTE = 10:1), monodisperse (PDI = 0.23) stable associates **7**/PVTE with an average hydrodynamic diameter 116 nm are formed (see Appendix A). There is a dramatic increase in the mean hydrodynamic diameter and PDI with increasing PVTE concentration. Both systems **7**/PVTE and **9**/PVTE do not form stable associates in aqueous and aqueous–alcoholic (H_2_O/C_2_H_5_OH = 100/1) solutions (1 × 10^−3^–1 × 10^−5^ M) (see Appendix A).

### 3.4. Study of Self-Assembly of a Three-Component System **7**/Flu/PVTE

It should be noted that despite the presence of the interaction for **9**/Flu and **9**/PVTE confirmed by UV–Vis spectroscopy, the formation of associates **9**/Flu and **9**/PVTE by the DLS method was not detected. However, in the case of the **7**/Flu and **7**/PVTE systems, the formation of stable associates occured, which was confirmed by UV–Vis method, fluorescence spectroscopy, and DLS. The results obtained indicate the possibility of the formation of ternary self-assembling systems **7**/Flu/PVTE, in which the pillar[5]arene acts as a molecule capsule for the drug compound, and the polymer acts as an outer protective shell (Figure 6a).

To confirm this hypothesis, it is necessary to check the interaction between PVTE and Flu. The UV–Vis spectroscopy and the dynamic light scattering methods showed no changes in the spectra of the PVTE/Flu system, and the formation of PVTE/Flu associates in the 50:1,10:1, 5:1, 2:1, 1:1, 1:2, 1:5, and 1:15 ratios (1 × 10^−3^–1 × 10^−5^ M) (see Appendix A). Next, the interaction of the **7**/Flu = 1:1 system with the PVTE polymer in the ratios **7**/Flu/PVTE = 1:1:0.1, 1:1:1, 1:1:5, 1:1:10 was studied by the method of DLS (1 × 10^−3^–1 × 10^−5^ M) in ethanol and aqueous–alcoholic solution (H_2_O/C_2_H_5_OH = 100/1). It turned out that the addition of small amounts of PVTE (1 × 10^−6^–5 × 10^−6^ M) to the system **7**/Flu = 1:1 (1 × 10^−5^ M) does not lead to a change in the average hydrodynamic diameter of the system itself **7**/Flu = 1:1 (Figure 6a, Appendix A), whereas adding a 10-fold excess of PVTE (1 × 10^−4^ M) stabilizes the system (ζ = −12.81 mV) **7**/Flu/PVTE (see Appendix A). The hydrodynamic diameter **7**/Flu/PVTE = 1:1:10 decreases to 48 nm with PDI = 0.16. Interestingly, the tendency for the formation of nano-sized associates **7**/Flu/PVTE = 1:1:10 persists, with a slight increase in the diameter particle and average PDI system, when the concentration of **7**/Flu = 1:1 (1 × 10^−3^–1 × 10^−4^ M) is increasing (see Appendix A).

As noted above, 7/Flu and 7/PVTE separately do not form associates in the H_2_O/C_2_H_5_OH = 100/1 system. However, the nanoscale associates remain (ζ = −34.12 mV), when a solution of the ternary system 7/Flu/PVTE = 1:1:10 in ethanol is transferred into water (Figure 6a, Appendix A). Attempts to form the 7/Flu/PVTE ternary system directly in H_2_O/C_2_H_5_OH resulted in a polydisperse system. The ability to form nanostructured 7/Flu/PVTE associates in the H_2_O/C_2_H_5_OH system was investigated by scanning electron microscopy (SEM). According to SEM data (Figure 6, Appendix A), PVTE polymer (Figure 6b, Appendix A) forms micron-sized dendritic aggregates. The 7/Flu/PVTE system (Figure 6c,d, Appendix A) is characterized by spherical associates with an average diameter of 66 nm. It should be noted that the 7/Flu system in H_2_O/C_2_H_5_OH represents formless aggregates of submicron sizes (Figure 6e, Appendix A). Thus, the preliminary self-assembly of the 7/Flu/PVTE system in ethanol followed by its transfer to water opens up the possibility of forming nanostructured associates with an incorporated drug. However, drug release processes are also of interest. One convenient way to detect the release of a potential drug is to track spectral changes when a nanostructured system is exposed to certain stimuli. Among these stimuli, one of the most promising is the response of the system to a change in pH. This is due to the pH values in late endosomes and/or lysosomes in tumor cells that differ from those in healthy tissues and equal to approximately 5.0 [67,68]. Thus, the methods of DLS and fluorescence spectroscopy were used to study the behavior of the 7/Flu/PVTE (1:1:10) system at different pH. System 7/Flu/PVTE was prepared in buffers at pH 2–9 by adding an alcohol solution of the system to the buffer in the ratio C_2_H_5_OH/buffer = 1/100.

The **3**/Flu/PVTE (1:1:10) system in a buffer at pH 9–7 (see Appendix A) remains stable (ζ = −31.22 mV—pH 9 and ζ = −33.14 mV—pH 7) and monodisperse (D = 83 nm—pH 9 and D = 84—pH 7) with a polydispersity index (PDI) = 0.12–0.15, according to the DLS results. The retention of encapsulated Flu in the **7**/Flu/PVTE system is confirmed by changes in the fluorescence spectra (Figure 6f). Thus, the position of the emission maximum does not change and a slight flare-up of fluorescence is observed at λ_max_ = 512 nm in the fluorescence spectra. There is a dramatic increase in PDI (see Appendix A) and particle diameter **7**/Flu/PVTE in going from pH 7 to pH 5–2. The **7**/Flu/PVTE system becomes polydisperse in the pH range 5–4. The highest values of PDI (0.55) and average hydrodynamic diameter (D = 737 nm) are at pH 2 (Figure 6a, see Appendix A). According to the data of fluorescence spectroscopy, from pH 7 to pH 5–2, a sharp flare-up of fluorescence is observed, which indicates a change in the environment of the dye at a pH close to **7** (Figure 6f). At the same time, it is well known that the quantum yield of fluorescein upon protonation in an acidic medium is minimal [69]. Thus, it is obvious that even at a low pH value, Flu is still isolated from the solvent [70], which is possible when it is inside pillararene. Probably, its destruction occurs with the release of the polymer and the dye encapsulated in pillararene as a result of protonation of the outer shell of the ternary complex consisting of PVTE.

## 4. Conclusions

Thus, for the first time, a new type of universal stimulus-sensitive system DDS based on decasubstituted macrocyclic structures—pillar[5]arenes and tetrazole-containing polymers—was demonstrated. Decasubstituted pillar[5]arenes 6 and 11, containing easily leaving tosylate and phthalimide fragments, were synthesized. Removal of tosylate and phthalimide protections yielded pillar[5]arenes 7–9 and 12, containing primary and tertiary amino groups in high yield. The ability of pillar[5]arenes 7 and 9 containing tertiary amine groups to interact with the tetrazole-containing polymer PVTE and the Flu dye was shown by UV–Vis and fluorescence spectroscopy. The DLS method showed the absence of associates in the 9/Flu and 9/PVTE systems. However, in the case of 7/Flu systems (1:1, C7/Flu = 1 × 10−5 M, D = 155 nm, PDI = 0.16) and 7/PVTE (10:1, C7 = 1 × 10−4 M, CPVTE = 1 × 10−5 M, D = 116 nm, PDI = 0.23), stable associates are formed. The possibility of forming a triple self-assembly system 7/Flu/PVTE was shown. The average hydrodynamic diameter 7/Flu/PVTE decreases to 48 nm with PDI = 0.16. The behavior of the 7/Flu/PVTE (1:1:10) system was studied at different pH by DLS and fluorescence spectroscopy. It was shown that the 7/Flu/PVTE system becomes polydisperse in going from pH 7 to pH 5-2. The results obtained open a broad range of possibilities for the development of new universal stimulus-responsive DDS containing nontoxic tetrazole-based polymers.

## Data Availability

Data is available upon the reasonable request from the corresponding author.

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
