# Peer review of "Towards Universal Stimuli-Responsive Drug Delivery Systems: Pillar[5]arenes Synthesis and Self-Assembly into Nanocontainers with Tetrazole Polymers"

_nanomaterials, 2021, doi:10.3390/nano11040947_

Round 1

Reviewer 1 Report

Manuscript titled ‘Towards universal stimuli-responsive Drug Delivery Systems based on the tetrazole-containing polymers: synthesis of pillar[5]arenes and their self-assembly into nanocontainers’ by Shurpik et al. submitted to Nanomaterials as a Research Article. The dataset is well presented; however, revision is needed to address the following comments.

  1. Please simplify the title, it is too long
  2. Abstract is poorly constructed: there is no introduction, no building story, no innovation and signification. There are some errors as well: ‘pH close to 5. stract text goes here.’ The abstract reads like it is just presenting the facts and results, it needs heavy revision and rephrasing.
  3. Line 29-30. The first line of introduction talks about supramolecular chemistry and responsive delivery, without giving any information about why it is needed. The paper is written only for specific readers?? The non-specific reader will be confused.
  4. Entire first para of introduction again does not build any story. It just randomly presents the facts without any link. It just assumes that reader will have in-depth knowledge of the subject. What is the application you are targeting? Drug delivery is a very vast subject.
  5. Please introduce para-cyclophanes, pillar[n]arenes.
  6. What is the novelty, significance and application of the current study? Please write it in simple terms, sprinkling it in abstract and conclusion. How it is different from current studies in this domain (as cited by authors). Specially references by Authors 17-22.
  7. Stimuli responsive drug delivery need and application is poorly described (Line 64-66). It is unclear what authors are trying to say/achieve.
  8. Section 2.1. Not sure why it is labelled General? The sentences are separate and there is no paragraph? This section needs to be written properly with full details like a full research paper. No details are present – just one small sentence for each instrument (what conditions, what analysis, parameters, etc. are missing).
  9. Line 258: Loosening of the cell membrane during drug transport can be detrimental to the cell. Authors to cite appropriate references and explain why.
  10. Line 261: In previous works, we showed [42], that polymers based on tetrazole in the presence of pillar[5]arene are capable of forming nanoscale associates. How is current study significantly different from Authors previous work. Please elaborate.
  11. Line 290: Macrocyclization now is the main method for obtaining substituted pillar[n]arenes. Please rephrase.

Author Response

1) Please simplify the title, it is too long.

Answer: The comments of the Reviewer were taken into account. The title was changed from “Towards universal stimuli-responsive Drug Delivery Systems based on the tetrazole-containing polymers: synthesis of pillar[5]arenes and their self-assembly into nanocontainers” to “Towards universal stimuli-responsive Drug Delivery Systems: pillar[5]arenes synthesis and self-assembly into nanocontainers with tetrazole polymers”

2) Abstract is poorly constructed: there is no introduction, no building story, no innovation and signification. There are some errors as well: ‘pH close to 5. stract text goes here.’ The abstract reads like it is just presenting the facts and results, it needs heavy revision and rephrasing.

Answer: The comments of the reviewer were taken into account.

The abstract was changed from:

“New decasubstituted pillar[5]arenes containing a large, good leaving tosylate and phthalimide groups were first synthesized and characterized. Pillar[5]arenes containing primary and tertiary amino groups were obtained with high yield by removing the tosylate and phthalimide protection. The ability to form an inclusion complex by pillar[5]arene containing tertiary amino groups with the fluorescein dye (Ka = 10614) and tetrazole fragments of the PVTE polymer (Ka = 2994) was shown. The triple system pillar[5]arene / fluorescein / polymer forms spherical associates with an average diameter of 66 nm in aqueous solutions. It was found according to the fluorescence spectroscopy data, a dramatically fluorescence enhancement in the pillar[5]arene / fluorescein / polymer system is observed with decreasing pH from neutral (pH = 7) to acidic (pH = 5). This indicates the destruction of associates and the release of the dye at the pH close to 5.”

to

“In this work, we have proposed a novel universal stimulus-sensitive nanosized polymer system based on decasubstituted macrocyclic structures - pillar[5]arenes and tetrazole-containing polymers. Decasubstituted pillar[5]arenes containing a large, good leaving tosylate and phthalimide groups were first synthesized and characterized. Pillar[5]arenes containing primary and tertiary amino groups, capable of interacting with tetrazole-containing polymers, were obtained with high yield by removing the tosylate and phthalimide protection. According to the fluorescence spectroscopy data, a dramatically fluorescence enhancement in the pillar[5]arene / fluorescein / polymer system is observed with decreasing pH from neutral (pH = 7) to acidic (pH = 5). This indicates the destruction of associates and the release of the dye at the pH close to 5. The presented results open up wide opportunities for the development of new universal stimulus-sensitive drug delivery systems containing macrocycles and non-toxic tetrazole-based polymers.”

3) Line 29-30. The first line of introduction talks about supramolecular chemistry and responsive delivery, without giving any information about why it is needed. The paper is written only for specific readers?? The non-specific reader will be confused.

Answer: The first paragraph of the introduction has been rewritten.

The text was added to the manuscript:

“In recent years, the pharmaceutical industry has increasing interest in the development and methods of introducing nanosystems in the treatment of various diseases by encapsulating drugs in biocompatible polymer matrices [1-3]. The resulting polymer-drug associates can change the pharmacokinetic properties and profile of the loaded drug after administration and provide a controlled and long-term effect of drugs on disease foci in comparison with the effect of the drug itself [1-3]. In addition, the polymer shell protects the loaded drug from premature biotransformation and can transport the drug to the focus of the disease practically without damage [1]. Water-soluble polymer systems occupy a special place among such drug delivery systems (DDS) [4, 5]. Various types of synthetic and natural polymer compositions (solid liquid nanoparticles, liposomes, etc.) are widely studied as promising DDS [6, 7]. However, polymer systems have a number of disadvantages, namely, an extremely developed spatial structure, weak receptor properties, which complicates their controlled interaction with drugs [8, 9].”

4) Entire first para of introduction again does not build any story. It just randomly presents the facts without any link. It just assumes that reader will have in-depth knowledge of the subject. What is the application you are targeting? Drug delivery is a very vast subject.

Answer: The first paragraph of the introduction has been rewritten.

5) Please introduce para-cyclophanes, pillar[n]arenes.

Answer: The comments of the reviewer were taken into account. The following text was added to the manuscript:

“Pillar[n]arenes are macrocyclic compounds in which fragments of substituted hydroquinones are interconnected by methylene bridges.”

6) What is the novelty, significance and application of the current study? Please write it in simple terms, sprinkling it in abstract and conclusion. How it is different from current studies in this domain (as cited by authors). Specially references by Authors 17-22..

Answer: The comments of the reviewer were taken into account.

The text was added to the manuscript:

“In this study, we propose a new type of universal stimulus-sensitive DDS system based on decasubstituted macrocyclic structures - pillar[5]arenes and tetrazole-containing polymers. The versatility of the system lies in the principle of step-by-step supramolecular self-assembly of DDS components. The macrocylic compound in this case will act as a link between the "protective shell" - a non-toxic water-soluble polymer and a drug. This is due to the presence of a macrocyclic cavity in pillar[5]arenes [17-22], which is involved in the formation of host-guest complexes with drugs of various structures [18, 21, 22]. Also, the introduction into the structure of macrocycles of substituents complementary to the tetrazole fragments of the polymer and sensitive to pH changes will facilitate the packing of tetrazole-containing polymers into nanosized associates [17, 19].”

7) Stimuli responsive drug delivery need and application is poorly described (Line 64-66). It is unclear what authors are trying to say/achieve.

Answer: The comments of the reviewer were taken into account. The text was added to the manuscript:

«One of the key conditions for creation of DDS based on water-soluble polymers is micellization of polymer systems in a wide pH range [32-34]. DDS based on polymer systems are of practical interest as nanocarriers for the encapsulation and controlled release of hydrophobic drugs. Such nanosized polymer systems are sensitive to the ambient pH due to the presence of charged ammonium fragments in their structure [33]. Tetrazole-containing polymer systems are ideal as components of pH-sensitive DDS. However, it should be noted that polymers based on PVT do not form stable nanosized aggregates in aqueous solutions [34]. In this regard, the formation of stable, stimulus-responsive nanoscale associates of PVT / pillararene capable of controlled drug release is an actual problem.»

[32] Atanase, L. I.; Riess, G. Micellization of pH-stimulable poly (2-vinylpyridine)-b-poly (ethylene oxide) copolymers and their complexation with anionic surfactants. J. Colloid Interface Sci. 2013, 395, 190-197.

[33] Iurciuc-Tincu, C. E.; Cretan, M. S.; Purcar, V.; Popa, M.; Daraba, O. M.; Atanase, L. I.; Ochiuz, L. Drug delivery system based on pH-sensitive biocompatible poly (2-vinyl pyridine)-b-poly (ethylene oxide) nanomicelles loaded with curcumin and 5-fluorouracil. Polymers. 2020, 12, 1450.

[34] Atanase, L. I.; Lerch, J. P.; Caprarescu, S.; Iurciuc, C. E.; Riess, G. Micellization of p H‐sensitive poly (butadiene)‐block‐poly (2 vinylpyridine)‐block‐poly (ethylene oxide) triblock copolymers: Complex formation with anionic surfactants.        J. Appl. Polym. Sci. 2017, 134, 45313-45321.

8) Section 2.1. Not sure why it is labelled General? The sentences are separate and there is no paragraph? This section needs to be written properly with full details like a full research paper. No details are present – just one small sentence for each instrument (what conditions, what analysis, parameters, etc. are missing).

Answer: Reviewer's comments were considered and manuscript were edited.

9) Line 258: Loosening of the cell membrane during drug transport can be detrimental to the cell. Authors to cite appropriate references and explain why.

Answer: The comments of the reviewer were taken into account. The text was added to the manuscript:

“Loosening of the cell membrane during drug transport can be detrimental to the cell. The interaction between water-soluble polymers and lipid membranes is mainly governed by the hydrophobic / hydrophilic balance as well as the polymer architecture [45]. Charged polymers (polyelectrolytes), such as polycarboxylates or polyamines, can adsorb on oppositely charged lipid bilayers, which can lead to membrane deformations. This manifests itself in an increase in the curvature of the membrane, which can lead to sealing or destruction of the membrane in the case of cationic polymers [45,46]. Membrane damage occurs through hole formation, and some cationic polymers / oligomers can move across the cell membrane and bind to intracellular DNA or RNA to prevent intracellular synthesis [46].”

[45] Schulz, M.; Olubummo, A.; Binder, W. H. Beyond the lipid-bilayer: interaction of polymers and nanoparticles with membranes. Soft Matter. 2012, 8, 4849-4864.

[46] Fu, L.; Wan, M.; Zhang, S.; Gao, L.; Fang, W. Polymyxin B loosens lipopolysaccharide bilayer but stiffens phospholipid bilayer. Biophys. J. 2020, 118, 138-150.

10) Line 261: In previous works, we showed [42], that polymers based on tetrazole in the presence of pillar[5]arene are capable of forming nanoscale associates. How is current study significantly different from Authors previous work. Please elaborate.

Answer: The comments of the reviewer were taken into account. The text was added to the manuscript:

“In previous works, we showed [42], that polymers based on tetrazole in the presence of pillar[5]arene are capable of forming nanoscale associates. In the work [42], uncharged pillar[5]arenes containing hydroxyl groups were used. We have shown that the macrocyclic platform plays the key role in the formation of supramolecular associates. The resulting nanosized associates turned out to be insensitive to pH changes. Therefore, in this work, it was decided to use pillar[5]arenes containing primary and tertiary amino groups capable of protonated in an acidic medium.”

11) Line 290: Macrocyclization now is the main method for obtaining substituted pillar[n]arenes. Please rephrase.

Answer: Fragment of the manuscript: "Macrocyclization now is the main method for obtaining substituted pillar[n]arenes" was changed to "The cyclization of O-substituted hydroquinones in the presence of paraformaldehyde is currently the main method for the preparation of substituted pillar[n]arenes."

Reviewer 2 Report

The paper entitled “Towards universal stimuli-responsive Drug Delivery Systems based on the tetrazole-containing polymers: synthesis of pillar[5]arenes and their self-assembly into nanocontainers” by Dmitriy Shurpik , Lyaysan Makhmutova , Konstantin Usachev , Daut Islamov , Olga Mostovaya , Anastasia Nazarova , Valeriy Kizhnyaev , Ivan Stoikov presents a procedure of synthesis and characterization of pillar[5]arenes. The ability to form an inclusion nanocomplexes with the selected actives and polymer was shown. Since the topic of the work is quite interesting and in the scope of the Journal, as well as based on the quality of the manuscript I recommend the publication of this paper.

There is only one suggestion concerning the conclusion paragraph. The present form is too long and repeats the results presented in the previous sections. Please, rewrite it, do not repeat results, conclusion should present short summary, what you learn from this work, and some future plans. 

Author Response

1) There is only one suggestion concerning the conclusion paragraph. The present form is too long and repeats the results presented in the previous sections. Please, rewrite it, do not repeat results, conclusion should present short summary, what you learn from this work, and some future plans.

Answer: Reviewer's comments were considered and сonclusions was reduced.

The text was added to the manuscript:

“Thus, for the first time, a new type of universal stimulus-sensitive system DDS based on decasubstituted macrocyclic structures - pillar[5]arenes and tetrazole-containing polymers was demonstrated. Decasubstituted pillar[5]arenes 6 and 11, containing easily leaving tosylate and phthalimide fragments, have been synthesized. Removal of tosylate and phthalimide protections yielded pillar[5]arenes 7-9 and 12, containing primary and tertiary amino groups in high yield. The ability of pillar[5]arenes 7 and 9 containing tertiary amine groups to interact with the tetrazole-containing polymer PVTE and the Flu dye was shown by UV-Vis and fluorescence spectroscopy. The DLS method showed the absence of associates in the 9/Flu and 9/PVTE systems. However, in the case of 7/Flu systems (1:1, С7/Flu = 1×10-5М, D = 155 nm, PDI = 0.16) and 7/PVTE (10:1, С7= 1×10-4 М, СPVTE = 1×10-5 M, D = 116 nm, PDI = 0.23), stable associates are formed. The possibility of forming a triple self-assembly system 7/Flu/PVTE was shown. The average hydrodynamic diameter 7/Flu/PVTE decreases to 48 nm with PDI = 0.16. The behavior of the 7/Flu/PVTE (1: 1: 10) system was studied at different pH by DLS and fluorescence spectroscopy. It was shown that the 7/Flu/PVTE system becomes polydisperse in going from pH 7 to pH 5-2. The results obtained open up wide possibilities for the development of new universal stimulus-responsive DDS containing non-toxic tetrazole-based polymers.”

Reviewer 3 Report

The paper submitted by Shurpik et al. is very interesting, original and describes a very well conducted study with a huge amount of data. The manuscript is clear, well written and the conclusions are supported by the results. Some minor corrections are needed, however, before its publication:

  1. Line 37: only amphiphilic copolymers can promote the solubilization of slightly water soluble drugs, therefore the authors must correct this sentence.
  2. The introduction section can be completed with several references concerning the investigation of pH-sensitive vinylic copolymers: https://doi.org/10.3390/polym12071450; https://doi.org/10.1002/app.45313; https://doi.org/10.1016/j.jcis.2012.12.058
  3. I think that fig S33 can be given in the main text.

Author Response

1) Line 37: only amphiphilic copolymers can promote the solubilization of slightly water soluble drugs, therefore the authors must correct this sentence.

Answer: Reviewer's comments were considered and this sentence was edited.

2 The introduction section can be completed with several references concerning the investigation of pH-sensitive vinylic copolymers: https://doi.org/10.3390/polym12071450; https://doi.org/10.1002/app.45313; https://doi.org/10.1016/j.jcis.2012.12.058.

Answer: Links have been added into the text of the manuscript.

2) I think that fig S33 can be given in the main text.

Answer: We believe that figure S33 is not suitable as an illustration in the main text of the manuscript and carries more technical information. The text of the manuscript contains all the necessary data and references to fig S33 in the ESl. 

Round 2

Reviewer 1 Report

Authors have addressed all comments. Article can now be accepted for publication.